materials science/chemical engineering

gas separation, membrane, poly(ether-*block*-amide) copolymer

**Authors for correspondence:**
Magdalena Malankowska
e-mail: magnal@unizar.es
Joaquín Coronas
e-mail: coronas@unizar.es

This article has been edited by the Royal Society of Chemistry, including the commissioning, peer review process and editorial aspects up to the point of acceptance.

# Poly(ether-*block*-amide) copolymer membrane for $CO_2/N_2$ separation: the influence of the casting solution concentration on its morphology, thermal properties and gas separation performance

Lidia Martínez-Izquierdo, Magdalena Malankowska, Javier Sánchez-Laínez, Carlos Téllez and Joaquín Coronas

Chemical and Environmental Engineering Department, Instituto de Nanociencia de Aragón (INA) and Instituto de Ciencia de Materiales de Aragón (ICMA), Universidad de Zaragoza-CSIC, 50018 Zaragoza, Spain

MM, 0000-0001-9595-0831

The present work is focused on the study of the effect that the casting solution concentration has on the morphology and gas separation performance of poly(ether-*block*-amide) copolymer membranes (Pebax® MH 1657). With this aim, three different concentrations of Pebax® MH 1657 in the casting solution (1, 3 and 5 wt%) were used to prepare dense membranes with a thickness of 40 μm. The morphology and thermal stability of all membranes were characterized by scanning electron microscopy, X-ray diffraction, differential scanning calorimetry, rotational viscometry and thermogravimetric analyses. An increase in crystallinity was notable when the amount of solvent in the Pebax® MH 1657 solution was higher, mainly related to the polymer chains arrangement and the solvent evaporation time. Such characteristic seemed to play a key role in the thermal degradation of the membranes, confirming that the most crystalline materials tend to be thermally more stable than those with lower crystallinity. To study the influence of their morphology and operating temperature on the $CO_2$

separation, gas separation tests were conducted with the gas mixture $CO_2/N_2$. Results indicated that a compromise must be found between the amount of solvent used to prepare the membranes and the crystallinity, in order to reach the best gas separation performance. In this study, the best performance was achieved with the membrane prepared from a 3 wt% casting solution, reaching at 35°C and under a feed pressure of 3 bar, a $CO_2$ permeability of 110 Barrer and a $CO_2/N_2$ selectivity of 36.

# 1. Introduction

Carbon dioxide is a final combustion product of carbon-containing fuels. It is generated in big quantities and emitted in the gaseous form in the case of industrial and energy production sites, transportation, building heating, etc. Such emission causes an increase in the $CO_2$ concentration in the atmosphere and contributes to the so-called climate change. $CO_2$ is a primary greenhouse gas and it is estimated that stationary $CO_2$ emissions are responsible for more than 60% of the overall $CO_2$ global emissions. To mitigate the effect of $CO_2$ in the atmosphere, its emission needs to be reduced by a substantial amount [1,2].

Membrane technology is, since the 1970s, one of the most studied techniques for the separation and sequestration of $CO_2$ from non-polar gases (such as $CO_2/N_2$, $H_2/CO_2$ and $CO_2/CH_4$ gas mixtures) due to its well-known advantages over the conventional methods, i.e. mechanical simplicity, easy to scale up, lower energy consumption and smaller footprints [3]. Lin & Freeman [4] reviewed the design strategies of membrane materials selection for the separation of $CO_2$ from gas mixtures. The introduction of polar groups with affinity to $CO_2$ is a promising method to raise $CO_2$/non-polar gases selectivity. Poly(ether-*block*-amide) (PEBA) copolymers are especially interesting due to the permeation selectivity of polar to non-polar gases. PEBA block copolymers are synthesized from polyoxyalkylene glycols (PEG or PTMG) and dicarboxylic acid terminated aliphatic polyamides (such as nylon-6 or nylon-12) [3]. Such block copolymers consist of soft (rubbery) and hard (glassy) segments that provide high gas permeability without the loss of selectivity and mechanical stability [5,6]. Nevertheless, in spite of its good properties, many efforts are still being made to improve even more its $CO_2$ separation performance, although the majority of them concern the incorporation of nanoparticles into the polymeric matrix or the synthesis of a composite membrane [7–10].

In the open literature, there are still few studies concerning the raw material, which should be taken more into account to choose the best conditions for further applications. The way in which membranes are prepared, i.e. solvent selection, solvent evaporation temperature, etc., is proved to have an effect on its morphology and hence on its gas separation performance [11,12]. Shao *et al.* [13] studied the influence of solvents on the morphology of 6FDA/PMDA–TMMDA copolyimide membranes. They found that those prepared with solvents possessing solubility parameters closer to that of the polymer had a better affinity to it, and thus, polymer chains mobility was higher, resulting in more crystalline structures and therefore in less permeable membranes. Karamouz *et al.* [12] studied the effect that the solvent evaporation temperature had on the membrane performance in gas separation. They found that the evaporation rate (higher when increasing the temperature) resulted in a more disordered phase at the top of the membrane, which led to more permeable and selective membranes.

As far as we are concerned, while these aforementioned parameters have already been studied for PEBA [11], there is no study about how the concentration in the casting solution affects the crystallinity and thermal properties of the polymer Pebax® MH 1657, although already studied by Ren *et al.* [5] for the gas separation performance. This is of great importance to find correlation between the casting conditions and the whole membrane performance, i.e. not only what limits the separation performance but what relates to other issues such as morphology, preparation reproducibility, long period stability, etc. The changes in the concentration of polymer in the casting solution directly affect the viscosity, which will lead to differences in the behaviour of membranes. As well as the solvent type, the viscosity will affect the evaporation time, and this will result in differences in crystallinity and hence in permeability and selectivity, as occurred in the studies of Shao *et al.* [13] and Karamouz *et al.* [12]. Therefore, with the aim of further optimizing the PEBA, in this work, we have carried out a study of the influence of the concentration of Pebax® MH 1657 in the casting solution on the corresponding gas separation membranes. In particular, the morphology, thermal properties and $CO_2$ separation performance of the PEBA membranes have been studied. With this aim, three different

casting solutions of PEBA (1, 3 and 5 wt%) have been prepared and the resulting membranes have been tested for gas separation using the $CO_2/N_2$ gas mixture.

# 2. Experimental methods

## 2.1. Materials

Polyether-*block*-amide, Pebax® MH 1657 (comprising 60 wt% polyethylene oxide (PEO) and 40 wt% aliphatic polyamide (PA6)) in the form of pellets was kindly provided by Arkema, France. The solvent, absolute ethanol, was purchased from Gilca, Spain. All gases used for the gas permeation tests were of research grade (greater than 99.9% pure) and supplied by Abelló Linde S.A., Spain. All gases and solvents were used as received.

## 2.2. Membrane preparation

Pebax® MH 1657 (1, 3 and 5 wt%) was dissolved in a mixture of ethanol and water (70/30 (v/v)) by stirring under reflux at 80°C for 2 h. Once dissolved and cooled down to room temperature, the PEBA solution was poured in a Petri dish and dried for 48 h in a top-drilled box under a solvent-saturated atmosphere at environmental conditions. Note that the same amount of polymer (0.2 g) has been used to prepare the three casting solutions and that the total weight of each solution changes due to the different amount of solvent used to prepare them. The amount of polymer was fixed to obtain membranes of above the same thickness (40 μm) to be easily compared. Afterwards, membranes were treated at 50°C in a vacuum oven for 6 h to evaporate the residual solvent retained in the films. For clarity, the synthesized membranes will be abbreviated as PEBA1, PEBA3 and PEBA5, corresponding to the numbers of the concentration of PEBA in the mixture of solvents. PEBA3 membranes have been introduced in an oven at 150°C during different periods of time (3 and 8 days) in order to check their thermal annealing. These last membranes will be abbreviated as PEBA3_3d and PEBA3_8d for the periods of 3 and 8 days, respectively.

## 2.3. Membrane characterization

Scanning electron microscopy (SEM) images of the membranes were obtained using an Inspect F50 model scanning electron microscope (FEI), operated at 10 kV. Cross-sections of membranes were prepared by freeze-fracturing after immersion in liquid $N_2$ and subsequently coated with Pt. Membranes were also characterized by X-ray diffraction (XRD) using a Panalytical Empyrean equipment with $CuK_\alpha$ radiation ($\lambda = 0.154$ nm), over the range of 5°–40° at a scan rate of 0.03° s$^{-1}$, to examine the *d*-spacing of the membranes. To calculate the membranes' crystallinity, differential scanning calorimetry (DSC) analyses were performed on a Mettler Toledo DSC822e. Samples (approx. 2 mg) placed in 70 μl aluminium pans were heated in 40 cm$^3$(STP) min$^{-1}$ of nitrogen flow from 25 to 250°C at a heating rate of 10°C min$^{-1}$. Viscosity tests were conducted with a SMART L Fungilab rotational viscometer. Casting solutions (approx. 15 ml) placed in an APM/B adapter were subjected to different rotational speeds (from 50 to 150 r.p.m.) at 20°C. Thermogravimetric analyses (TGA) and differential thermogravimetry (DTG) were carried out using a Mettler Toledo TGA/STDA 851e. Small pieces of membranes (approx. 3 mg) placed in 70 μl alumina pans were heated under an air flow (40 ml min$^{-1}$) from 35 to 700°C at a heating rate of 5, 10, 15 and 20°C min$^{-1}$.

## 2.4. Gas permeation measurements

The separation of the $CO_2/N_2$ mixture was performed in the experimental system that is schematically presented in scheme 1. The membranes were cut and placed in a module consisting of two stainless steel pieces and a 316LSS macroporous disc support (Mott Co.) with a 20 μm nominal pore size. Membranes, 2.12 cm$^2$ in area, were gripped inside with Viton O-rings. To control the temperature of the experiment, which has an effect on gas separation, the permeation module was placed in a UNE 200 Memmert oven. Gas separation measurements were carried out by feeding the post-combustion gaseous mixture of $CO_2/N_2$ (15/85 cm$^3$(STP) min$^{-1}$) at an operating pressure of 3 bar and various temperatures (25, 35 and 50°C) to the feed side, controlled by two mass-flow controllers (Alicat Scientific, MC-100CCM-D). The permeate side of the membrane was swept with 2 cm$^3$(STP) min$^{-1}$ of He, at atmospheric pressure

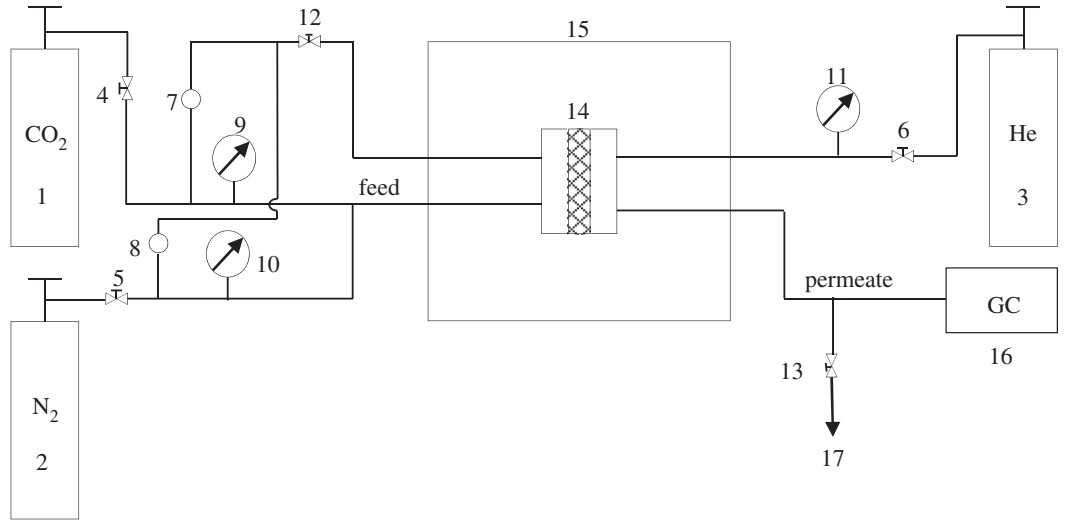

**Scheme 1.** Gas permeation experimental system. 1–3, gas cylinders; 4–6, ball valves; 7, 8, purge; 9–11, pressure gauges; 12, 13, needle valves; 14, permeation module; 15, oven; 16, gas chromatograph; 17, residue.

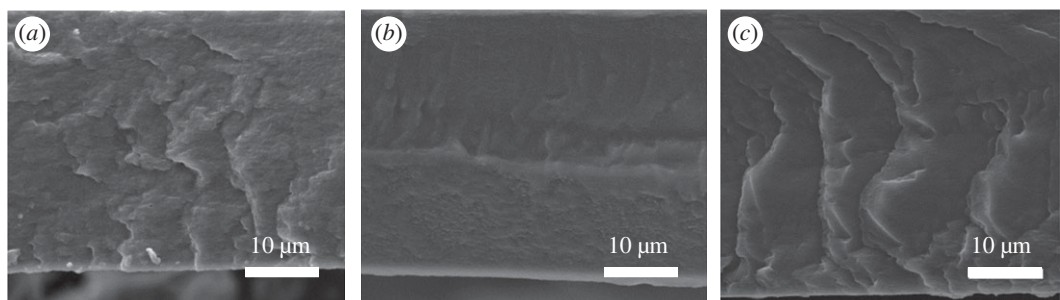

**Figure 1.** SEM images of dense PEBA membranes: PEBA1 (*a*), PEBA3 (*b*) and PEBA5 (*c*).

(approx. 1 bar) (Alicat Scientific, MC-5CCM-D). Concentrations of $N_2$ and $CO_2$ in the outgoing streams were analysed online by an Agilent 3000A micro-gas chromatograph. Permeability was calculated in Barrer ($10^{-10}$ cm$^3$(STP) cm cm$^{-2}$ s$^{-1}$ cm Hg$^{-1}$), once the steady state of the exit stream was reached (at least after 3 h). The separation selectivity was calculated as the ratio of permeabilities.

# 3. Results and discussion

## 3.1. Membrane characterization

Dense membranes with no defects usually follow the solution-diffusion model. This model assumes that no pores exist in the membrane, and thus, species are separated based on their solubility and diffusivity through the membrane, instead of molecular sieving [14]. Figure 1 shows the cross-sectional images of the three different membranes prepared in this work. For the three casting solution concentrations, the SEM images confirm the defect-free morphology of the PEBA membranes, without the existence of porosity or pinholes, which suggests that gases are transported following the solution-diffusion mechanism.

Figure 2 shows the XRD spectra of the PEBA membranes. Poly(ether-*block*-amide) copolymers are semi-crystalline polymers which consist of both amorphous and crystalline PEO and PA phases. In particular, Pebax® MH 1657 possesses two crystalline characteristic peaks. The first peak appears at a $2\theta$ value of 20.0°, attributed to the $\alpha$ crystalline phase of PA6, the most probable and stable phase presented in this polymer [15], corresponding to a $d$-spacing of 0.44 nm, and the other one at a $2\theta$ value of 23.8°, mainly associated with the less bulkier PEO segments [16], and to a molecular distance of 0.37 nm (see electronic supplementary material, equation S1). The amorphous region in these membranes comprises the incidence angle interval from 12.0° to 27.0°. Differences or displacements of crystalline peaks associated with the different concentration of PEBA used for the preparation of each membrane were not

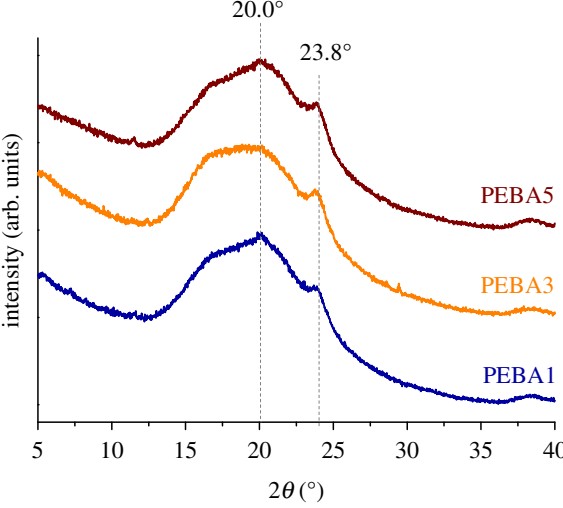

**Figure 2.** XRD patterns of PEBA dense membranes.

**Table 1.** Melting temperatures, crystallinity, maximum degradation temperatures and apparent activation energies for degradation of the PEBA membranes extracted from DSC and TGA analyses.

|  | $T_m$ PEO (°C) | $T_m$ PA (°C) | $X_c$ PEO (%) | $X_c$ PA (%) | $T^a_{max}$ (°C) | $T^b_{max}$ (°C) | $E^c_a$ (kJ mol$^{-1}$) | $E^d_a$ (kJ mol$^{-1}$) |
|---|---|---|---|---|---|---|---|---|
| PEBA1 | 14 | 206 | 15 | 6 | 414 | 521 | 277 | 274 |
| PEBA3 | 14 | 204 | 9 | 3 | 420 | 515 | 263 | 261 |
| PEBA5 | 12 | 203 | 8 | 3 | 418 | 511 | 218 | 218 |

[a]Thermal decomposition at a heating rate of 10°C min$^{-1}$.
[b]Oxidation step at a heating rate of 10°C min$^{-1}$.
[c]Calculated with the Kissinger equation.
[d]With the Ozawa method.

appreciated, so it can be stated that the three samples tested possessed similar molecular distances between their polymeric chains, in principle not being a crucial factor for gas permeabilities.

XRD analyses have been also carried out for the membranes subjected to thermal annealing, in order to analyse how crystallinity is affected by post-treatment at high temperature. Electronic supplementary material, figure S1 shows a comparison of the XRD diffraction peaks of the membranes with and without thermal treatment. As expected, the peak at 23.8° becomes more intense as days of treatment increase, suggesting a higher crystallinity. Despite this fact, the membranes acquired a toasted colour after each thermal treatment (electronic supplementary material, figure S2), revealing that a partial degradation has taken place during the heating. This fact was corroborated by TGA analyses (see Apparent activation energy section).

From the results obtained by DSC, it was possible to evaluate the crystallinity of the membranes without post-treatment (table 1). Figure 3 shows the thermograms of PEBA at the three tested concentrations. In this figure, two different melting peaks can be observed corresponding to the soft (PEO) and the hard (PA6) segments. The melting temperature of both segments was around 14°C for the PEO and 204°C for the PA, being in coherence with those reported previously in the literature [17]. The slight decrease in the melting temperature of both segments suggests that the crystallinity of samples becomes lower as the PEBA concentration in the casting solution increases. This fact means that the crystalline regions in the membrane decreased when the amount of solvent in the PEBA solution was lower, as previously reported with analogous polymers [18]. Crystallinity data collected in table 1 verify this phenomenon. The higher crystallinity of PEBA when decreasing its concentration could be explained taking into account the time required to completely evaporate the solvent. In fact, for the same amount of polymer, the higher the quantity of solvent, the longer the evaporation time

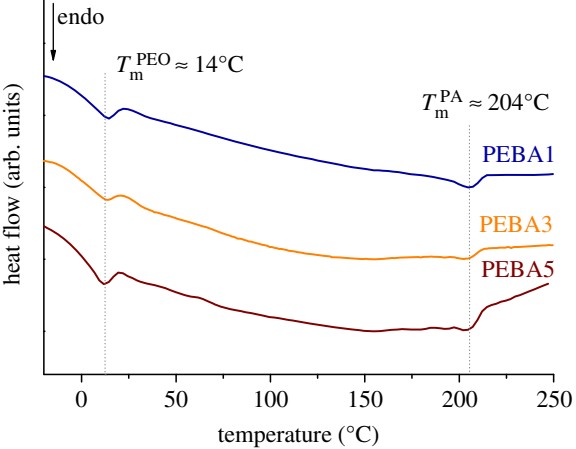

**Figure 3.** DSC thermograms of PEBA dense membranes.

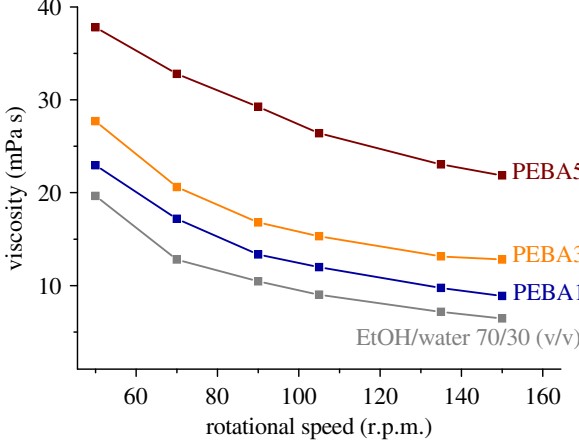

**Figure 4.** Viscosity of the three casting solutions (1, 3 and 5 wt%) and the EtOH/water (70/30 v/v) solvent at different rotational speeds.

becomes, and hence, the polymeric chains have more time to rearrange and form more organized crystalline structures [19].

Viscosity tests were conducted in order to corroborate the influence of the PEBA concentration on the viscosity of the casting solutions and its possible correlation with the evaporation time and crystallinity. Figure 4 compares, at different rotational speeds, the viscosity of the three casting solutions with that of the solvent. For all the solutions tested, the viscosity decresed with the increment of the rotational speed, which means that these solutions behave as non-Newtonian pseudoplastic fluids. As expected, an increase in viscosity with the increment of PEBA concentration can be clearly observed. Besides, while for the casting solutions prepared with 1 and 3 wt% of the polymer, the viscosity seems to follow the same tendency, a major increment can be observed in the case of the one with 5 wt% of PEBA. This fact suggests that the gas separation performance of the membrane prepared with 5 wt% casting solution may differ from the others. This statement will be confirmed in the Gas permeation measurements section (see figures 5 and 6).

## 3.2. Apparent activation energy

The thermal stability of the prepared membranes was studied by TGA and DTG analyses. Figure 7 shows the thermograms obtained for the membranes tested in this work (with and without thermal post-treatment). As observed in figure 7a and collected in table 1, the Pebax® MH 1657 is thermally stable up to 360°C (the temperature at which samples start to lose weight), and they reach their maximum degradation at 417°C. A second degradation step can also be observed related to the oxidation mechanism, in which the combustion of aromatic compounds and residues of the thermal degradation

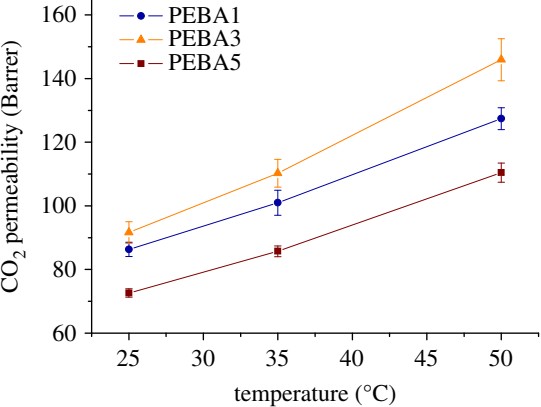

**Figure 5.** Comparison of $CO_2$ permeability at different temperatures (25, 35 and 50℃) and 3 bar feed pressure for the three concentrations of Pebax® MH1657 tested in this work. Error bars come from the testing of at least three different membrane samples.

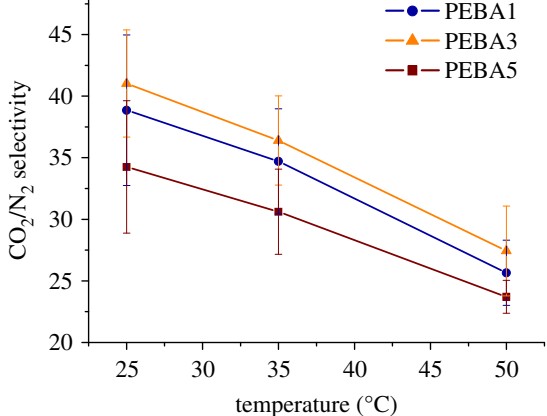

**Figure 6.** Comparison of $CO_2/N_2$ selectivity at different temperatures (25, 35 and 50℃) and 3 bar of feed pressure for the three concentrations tested.

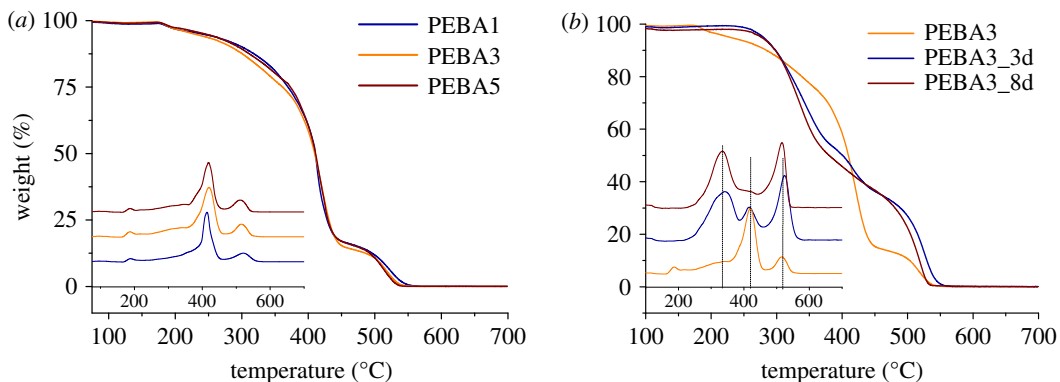

**Figure 7.** TGA and DTG curves of Pebax® MH 1657 at different concentrations (1, 3 and 5 wt%) (a) and with thermal annealing at 150℃ for different periods of time (0, 3 and 8 days) (b), oxidized in air atmosphere at 10℃ min$^{-1}$.

occurs [20]. This oxidative stage begins at 464℃, reaching its maximum peak at 516℃. Changes in the maximum degradation temperature associated with the different concentration of PEBA in the casting solution cannot be appreciated. With the data collected from the experiments carried out at different heating rates (electronic supplementary material, figures S3a–c), it was possible to calculate the apparent activation energy (table 1) with the Kissinger and Ozawa integral methods (electronic supplementary material, equations S4 and S5).

A drop in the apparent activation energy was noted when the concentration of PEBA in the casting solution increased. Such fall could be related to the reduction in the membrane crystallinity, which means more labile polymer chains (since chain mobility is higher). This way, the energy required to degrade the polymer decreases. The weaker polymeric chain interaction (inter- and intra-bonding between chains) renders a reduction in the thermal stability of membranes [21]. This result suggests that the reduction in the polymer concentration, and thus in the viscosity of the polymer solution, may facilitate to some extent the interaction between the polymer chains, helping them to reach a more favourable orientation to maximize the polymer–polymer interactions. On the contrary, an increase in the viscosity of the polymer solution would hinder the polymer chain interactions, decreasing the crystallinity of the final solid polymer. It will be shown in the next section that the differences in crystallinity affect the permeation performance.

As mentioned before, TGA were also carried out with the post-treated membranes in order to verify their partial degradation. As depicted in figure 7b and collected in electronic supplementary material, table S1, the membranes subjected to thermal treatment began to lose weight at lower temperatures (240°C). Furthermore, the final weight loss associated with the aromatic compounds (the peak at approx. 520°C) becomes higher, which indicates that the samples have lost part of its lineal compounds during the treatment. To aid in the corroboration of this statement of partial degradation, values of apparent activation energy have been calculated for these membranes (electronic supplementary material, figure S4a and b and table S1). Results indicate that the membranes subjected to thermal annealing have lower apparent activation energies than the ones without post-treatment, meaning that the samples have lost part of their thermal stability.

## 3.3. Gas permeation measurements

Gas permeation measurements were conducted for the post-combustion gaseous mixture ($CO_2/N_2$, (15/85)) under a feed presure of 3 bar and different temperatures (25, 35 and 50°C), in order to study the dependence of $CO_2$ permeability with the operating temperature. Figure 5 shows the effect of the operating temperature on PEBA membranes. It can be observed that for the three casting solution concentrations tested, the $CO_2$ permeability increased when the temperature raised, due to the thermal activation process. To study this behaviour more in depth, an Arrhenius modified model was applied to the permeability data as follows [22]:

$$P = P_0 \cdot \exp^{(-E_\mathrm{p}/R \cdot T)}, \tag{3.1}$$

where $P$ is the gas permeability ($CO_2$ or $N_2$ in this work) in Barrer, $P_0$ is the pre-exponential factor, $E_\mathrm{p}$ is the permeation activation energy in J mol$^{-1}$, $R$ is the ideal gas constant 8.314 J mol$^{-1}$ K$^{-1}$ and $T$ is the operating temperature in K. Based on this equation, the activation energy will be higher for less permeable gases [23]. In this study, the $E_\mathrm{p}$ for $CO_2$ (the most permeable gas of the mixture) was lower than that for the $N_2$ (listed in table 2), corroborating the previous statement. Furthermore, values of activation energy for permation were very similar to those found in the literature for this polymer (13.3 kJ mol$^{-1}$ for $CO_2$ and 30.4 kJ mol$^{-1}$ for $N_2$) [24]. Besides, the highest activation energies for both $CO_2$ and $N_2$ permeations were those of PEBA3 (14.9 and 28.0 kJ mol$^{-1}$ for $CO_2$ and $N_2$, respectively), being this sample which provided the highest permeability and selectivity (146 Barrer of $CO_2$ with a $CO_2/N_2$ selectivity of 27 at 50°C), although the three membranes tested possessed comparable values at the same temperature. A comparison of the permeability of $CO_2$ for the three membranes prepared is depicted in figure 5. As shown in this figure, the highest $CO_2$ permeabilities were those of PEBA3 (92, 110 and 146 Barrer at 25, 35 and 50°C, respectively), whereas PEBA5 provided the lowest flow values (73, 86 and 110 Barrer of $CO_2$ at 25, 35 and 50°C, respectively). Usually, crystalline polymers tend to be less permeable than the amorphous ones, since hard segments in semi-crystalline polymers block the movement of gas molecules through the membrane, due to the inflexibility of the chains [25]. Based on this statement, PEBA5 was expected to be the sample with the highest gas permeability in this study because it was the membrane with the lowest crystallinity, according to DSC analyses (table 1 and figure 3), followed by PEBA3 and PEBA1. Conversely, PEBA5 did not follow this behavior. The explanation for such a decrease in permeability may be linked to the solvent evaporation time. As aforementioned, for the same amount of polymer, PEBA5 required less time to evaporate the solvent, leading to a more entangled structure, because the polymer chains do not have enough time to reorder. Furthermore, the interactions between macromolecules increase when the polymer solution is more concentrated due to the rise of viscosity, which also leads to chain entanglement and network formation [26]. Such entanglement could be acting as a barrier for the carbon dioxide to

**Table 2.** Gas permeation properties of PEBA dense membranes tested at different operating temperatures (25, 35 and 50°C) and under a feed pressure of 3 bar.

| | temperature (°C) | $CO_2$ permeability (Barrer) | $CO_2/N_2$ selectivity | $E_p$ $CO_2$ (kJ mol$^{-1}$) | $E_p$ $N_2$ (kJ mol$^{-1}$) |
|---|---|---|---|---|---|
| PEBA1 | 25 | 86 ± 2 | 39 ± 6 | 12.5 | 27.0 |
| | 35 | 100 ± 4 | 35 ± 4 | | |
| | 50 | 127 ± 3 | 26 ± 3 | | |
| PEBA3 | 25 | 92 ± 4 | 41 ± 4 | 14.9 | 28.0 |
| | 35 | 110 ± 4 | 36 ± 4 | | |
| | 50 | 146 ± 7 | 27 ± 4 | | |
| PEBA5 | 25 | 73 ± 1 | 34 ± 5 | 13.5 | 25.1 |
| | 35 | 86 ± 2 | 31 ± 4 | | |
| | 50 | 110 ± 3 | 24 ± 1 | | |

diffuse through the membrane, reducing the gas permeability. This effect of entanglement has been previously reported by Isanejad *et al*. [11]. In their case, differences of crystallinity were related to the volatility of the solvent used to dissolve the PEBA. Such volatility implied that the time required to completely evaporate the solvent was distinct and so was the separation performance of membranes. In fact, they found that the most crystalline membranes were those with the highest selectivity and lowest permeability, in agreement with our findings.

Figure 6 depicts the effect that the operating temperature has on the $CO_2/N_2$ selectivity and compares the three membranes studied. While as shown above, the gas permeability increased when raising the working temperature (figure 5), the $CO_2/N_2$ selectivity became lower, hence following the Robeson trade-off relationship between permeability and selectivity [27]. In any event, activation energy values are always higher for the slowest permeating compound in the mixture ($N_2$), which is consistent with the decrease in selectivity observed as a function of temperature. Selectivity and permeability data obtained from gas chromatography tests are collected in table 2. When comparing the selectivities achieved for each sample, PEBA3 can be considered the membrane with the highest separation capacity, independent of the operating temperature (with 41 as the highest selectivity at 25°C). This fact means that this membrane was able to differentiate in a better way between both gas molecules, thus reaching a slightly higher separation capacity. Again, PEBA5 was found to be the sample with the lowest values (with a $CO_2/N_2$ selectivity of 34 at 25°C), whereas PEBA1 selectivity performance was similar to that of PEBA3 (39 at 25°C). Comparing the permeability and selectivity values presented in table 2, it can be observed that both gas selectivity and permeability are similarly influenced by the changes in the PEBA concentration in the casting solution. On the other hand, these two parameters (selectivity and permeability) are also influenced by the operating temperature. Taking PEBA1 as an example, permeability increased 48%, whereas the $CO_2/N_2$ selectivity decreased 52%, when increasing the temperature from 25 to 50°C.

# 4. Conclusion

Dense Pebax® MH 1657 membranes, with a thickness of 40 μm, were successfully prepared varying the polymer concentration in the 70/30 (v/v) ethanol/water solvent mixture. As predicted, membranes showed different behaviours depending on the PEBA concentration used to prepare the casting solution. The sample with the lowest concentration (1 wt%), and thus, the highest percentage of solvent, resulted in the most crystalline film. This characteristic was principally attributed to the polymer chain interactions established in the solvent solution (due to the changes in the solution viscosity) and the evaporation time, which seemed to be an important factor in the fabrication of organized structures. Crystallinity, besides, played a key role in the thermal degradation, the most stable membranes being those with the highest crystalline domain, indicative of the rigidity of the polymer chains. Apparent activation energies aided in confirming this behaviour. The PEBA membrane obtained from the most diluted polymer solution (1 wt%) was found to be the most

crystalline film (15% and 6% related to the crystalline PEO and PA segments, respectively), and therefore, its apparent activation energy (calculated applying two different integral methods) was also the highest (*ca* 275 kJ mol$^{-1}$). Thermal annealing has been demonstrated by treating the PEBA3 at 150°C and different periods of time (3 and 8 days). In contrast with the samples not treated, although the crystallinity increases after the thermal treatment, membranes are partially degraded, and hence, they start to lose weight at lower temperatures.

The polymer crystallinity decreased with the polymer concentration (tested at 1, 3 and 5 wt%) in the casting solution. However, from the point of view of the $CO_2$ permeability and $CO_2/N_2$ separation selectivity, a trade-off was reached between crystallinity and separation properties, 3 wt% being the most suitable polymer concentration studied. The operating temperature (25, 35 and 50°C) exerted an important influence in the permeability and selectivity of the membranes. The activation energy values of permeation obtained after applying the Arrhenius equation to the collected data were similar to those found in the literature (*ca* 13 and 26 kJ mol$^{-1}$ for $CO_2$ and $N_2$, respectively) and higher for the less permeable gas ($N_2$ in this study), consistent as expected with the decrease in $CO_2/N_2$ selectivity as a function of temperature. In summary, the membranes prepared from a 3 wt% PEBA solution were found to be the most permeable and selective membranes, independent of the operating temperature.

Data accessibility. The data are provided as electronic supplementary material. It includes the theoretical background of crystallinity and apparent activation energy calculations as well as additional XRD results of thermal annealing.

Authors' contributions. L.M.-I. carried out the laboratory work, participated in the membrane characterization, performed the gas permeation measurements and data analysis as well as wrote the manuscript; M.M. participated in the design of the experiments, participated in the membrane characterization, performed XRD analysis and critically revised the manuscript; J.S.-L. conceived the study, coordinated the study and revised the manuscript; C.T. participated in the design of the study and revised the manuscript; and J.C. designed the study, coordinated the study and revised the manuscript. All authors gave final approval for publication and agreed to be held accountable for the work performed therein.

Competing interests. We declare we have no competing interests.

Funding. This project has received funding from the European Union's Horizon 2020 research and innovation programme under grant agreement no. 760944 (MEMBER project). Also, financial support from the Spanish Ministry of Science, Innovation and Universities and FEDER (MAT2016-77290-R), the Aragón Government (T43-17R) and the ESF is gratefully acknowledged.

Acknowledgements. All the microscopy work was done in the Laboratorio de Microscopías Avanzadas at the Instituto de Nanociencia de Aragón (LMA-INA). L.M.-I. thanks the Aragón Government (DGA) and J.S.-L. thanks the Spanish Education Ministry Program FPU2014 for their PhD grants. Finally, the authors acknowledge the use of the Servicio General de Apoyo a la Investigación-SAI (Universidad de Zaragoza).

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
