## [Reviewer comments · Royal Society Open Science]

Review History

RSOS-190866.R0 (Original submission)

Review form: Reviewer 1

Is the manuscript scientifically sound in its present form?

No

Are the interpretations and conclusions justified by the results?

No

Is the language acceptable?

No

Is it clear how to access all supporting data?

No

Do you have any ethical concerns with this paper?

No

Have you any concerns about statistical analyses in this paper?

No

Recommendation?

Reject

Comments to the Author(s)

This manuscript describes the fabrication of polyether-block-amide (PEBA) membranes for CO₂ separation. The authors prepared the PEBA membranes by varying the polymer concentration in the casting solution (1, 3, 5 wt.%) to investigate its influence on the morphology, thermal property, and gas separation performance. It was found that the amount of polymer in the casting solution affected the crystallinity and thermal stability of prepared membranes and the optimal content of PEBA was 3.0 wt.% because it showed the highest CO₂ permeability and CO₂/N₂ selectivity due to the lowest crystallinity without the PEBA chain entanglement. However, I cannot accept this manuscript because the effect of polymer concentration in the casting solution on the properties of the membrane has already been reported in many previous studies and any clear explanation on the separation behavior of the prepared membranes according to the mainly investigated research results (morphology, thermal property) is not possible. Therefore, this manuscript is not suited to be published in 'Royal Society Open Science'.

Review form: Reviewer 2

Is the manuscript scientifically sound in its present form?

Yes

Are the interpretations and conclusions justified by the results?

Yes

Is the language acceptable?

Yes

Do you have any ethical concerns with this paper?

No

Recommendation?

Major revision is needed (please make suggestions in comments)

Comments to the Author(s)

Comments on RSOS-190866

The present study reported the effect of polymer concentration on the morphology, thermal properties and CO₂/N₂ separation performance of PEBA membrane. The topic of CO₂ separation is of general interests to the membrane society, and PEBA membrane is a promising candidate for CO₂ separation. Overall the layout of the manuscript is all right, and some results might be useful even though the manuscript suffered from insufficient novelty. Major revision is recommended before it is publishable, and specific comments are listed below:

1. Please specify which gas separation process was studied in the title.

2. Avoid using uncommon abbreviation of Pebax MH 1657 in the abstract. Full name is recommended.
3. A major weakness of the manuscript is that authors failed to justify the necessity to study the effect of polymer concentration, and missed other important preparation conditions, such as solvent type, solvent evaporation temperature.
4. Membrane preparation section: please indicate how much weight of solution was poured, and how to control the membrane thickness.
5. Some figure numbers in the manuscript were wrong.
6. Authors are recommended to further explain why evaporation time will have a effect on the reorder of the polymer chain and why the chain order would affect the permeability.
7. Authors stated that “comparing the permeability and selectivity values presented in Table 2, ... through the membrane”. The comparison was unfair and the conclusion was not solid. Taking PEBA1 for example, the permeability increased 48%, whereas the selectivity decreased 52% when increasing temp. from 25 to 50C. The gas selectivity is more influenced than the permeability.
8. Error bar for the figure of CO₂/N₂ selectivity was missing.

Review form: Reviewer 3

Is the manuscript scientifically sound in its present form?

Yes

Are the interpretations and conclusions justified by the results?

Yes

Is the language acceptable?

Yes

Is it clear how to access all supporting data?

No

Do you have any ethical concerns with this paper?

No

Have you any concerns about statistical analyses in this paper?

No

Recommendation?

Accept with minor revision (please list in comments)

Comments to the Author(s)

The manuscript describes the effect of concentration of polymer solution on the morphology and gas permeation properties of PEBA membranes. Authors have studied three different concentrations and the membranes were characterized by SEM, XRD and thermal analysis. Authors use 1, 3, 5 wt% of polymer concentration. Membrane cast using 3wt% polymer shows optimum properties. The work is a systematic study and it will be a useful contribution to the literature in this area. The manuscript is recommended for publication in Royal Society Open Science after minor corrections are applied. Specific comments are given below.

- 1) The figure numbers should be checked carefully.
- 2) English proof reading is recommended.

Decision letter (RSOS-190866.R0)

16-Jul-2019

Dear Dr Malankowska:

Title: Polyether-block-amide membrane: The influence of casting solution concentration on its morphology, thermal properties and gas separation performance
Manuscript ID: RSOS-190866

The editor assigned to your manuscript has now received comments from reviewers. We would like you to revise your paper in accordance with the referee and Subject Editor suggestions which can be found below (not including confidential reports to the Editor). Please note this decision does not guarantee eventual acceptance.

Please submit your revised paper before 08-Aug-2019. Please note that the revision deadline will expire at 00.00am on this date. If we do not hear from you within this time then it will be assumed that the paper has been withdrawn. In exceptional circumstances, extensions may be possible if agreed with the Editorial Office in advance. We do not allow multiple rounds of revision so we urge you to make every effort to fully address all of the comments at this stage. If deemed necessary by the Editors, your manuscript will be sent back to one or more of the original reviewers for assessment. If the original reviewers are not available we may invite new reviewers.

Please also include the following statements alongside the other end statements. As we cannot publish your manuscript without these end statements included, if you feel that a given heading is not relevant to your paper, please nevertheless include the heading and explicitly state that it is not relevant to your work.

- Funding statement

Please include a funding section after your main text which lists the source of funding for each author.

Yours sincerely,
Dr Laura Smith

Publishing Editor, Journals

RSC Associate Editor:

Comments to the Author:

After conflicting recommendations were received from the first two referees, a third reviewer was invited. On balance, we invite you to complete the revisions requested.

RSC Subject Editor:

Comments to the Author:

(There are no comments.)

Reviewers' Comments to Author:

Reviewer: 1

Comments to the Author(s)

This manuscript describes the fabrication of polyether-block-amide (PEBA) membranes for CO₂ separation. The authors prepared the PEBA membranes by varying the polymer concentration in the casting solution (1, 3, 5 wt.%) to investigate its influence on the morphology, thermal property, and gas separation performance. It was found that the amount of polymer in the casting solution affected the crystallinity and thermal stability of prepared membranes and the optimal content of PEBA was 3.0 wt.% because it showed the highest CO₂ permeability and CO₂/N₂ selectivity due to the lowest crystallinity without the PEBA chain entanglement. However, I cannot accept this manuscript because the effect of polymer concentration in the casting solution on the properties of the membrane has already been reported in many previous studies and any clear explanation on the separation behavior of the prepared membranes according to the mainly investigated research results (morphology, thermal property) is not possible. Therefore, this manuscript is not suited to be published in 'Royal Society Open Science'.

Reviewer: 2

Comments to the Author(s)

Comments on RSOS-190866

The present study reported the effect of polymer concentration on the morphology, thermal properties and CO₂/N₂ separation performance of PEBA membrane. The topic of CO₂ separation is of general interests to the membrane society, and PEBA membrane is a promising candidate for CO₂ separation. Overall the layout of the manuscript is all right, and some results might be useful even though the manuscript suffered from insufficient novelty. Major revision is recommended before it is publishable, and specific comments are listed below:

1. Please specify which gas separation process was studied in the title.
2. Avoid using uncommon abbreviation of Pebax MH 1657 in the abstract. Full name is recommended.
3. A major weakness of the manuscript is that authors failed to justify the necessity to study the effect of polymer concentration, and missed other important preparation conditions, such as solvent type, solvent evaporation temperature.
4. Membrane preparation section: please indicate how much weight of solution was poured, and how to control the membrane thickness.
5. Some figure numbers in the manuscript were wrong.
6. Authors are recommended to further explain why evaporation time will have a effect on the reorder of the polymer chain and why the chain order would affect the permeability.
7. Authors stated that “comparing the permeability and selectivity values presented in Table 2, ... through the membrane”. The comparison was unfair and the conclusion was not solid. Taking PEBA1 for example, the permeability increased 48%, whereas the selectivity decreased 52% when increasing temp. from 25 to 50C. The gas selectivity is more influenced than the permeability.
8. Error bar for the figure of CO₂/N₂ selectivity was missing.

Reviewer: 3

Comments to the Author(s)

The manuscript describes the effect of concentration of polymer solution on the morphology and gas permeation properties of PEBA membranes. Authors have studied three different concentrations and the membranes were characterized by SEM, XRD and thermal analysis. Authors use 1, 3, 5 wt% of polymer concentration. Membrane cast using 3wt% polymer shows optimum properties. The work is a systematic study and it will be a useful contribution to the literature in this area. The manuscript is recommended for publication in Royal Society Open Science after minor corrections are applied. Specific comments are given below.

- 1) The figure numbers should be checked carefully.
- 2) English proof reading is recommended.

Author's Response to Decision Letter for (RSOS-190866.R0)

See Appendix A.

RSOS-190866.R1 (Revision)

Review form: Reviewer 2

Is the manuscript scientifically sound in its present form?

Yes

Are the interpretations and conclusions justified by the results?

Yes

Is the language acceptable?

Yes

Do you have any ethical concerns with this paper?

No

Have you any concerns about statistical analyses in this paper?

No

Recommendation?

Accept as is

Comments to the Author(s)

I have carefully reviewed the responses, and my concerns have been adequately addressed.

Decision letter (RSOS-190866.R1)

12-Aug-2019

Dear Dr Malankowska:

Title: Polyether-block-amide membrane: The influence of casting solution concentration on its morphology, thermal properties and gas separation performance
Manuscript ID: RSOS-190866.R1

It is a pleasure to accept your manuscript in its current form for publication in Royal Society Open Science. The chemistry content of Royal Society Open Science is published in collaboration with the Royal Society of Chemistry.

RSC Associate Editor:
Comments to the Author:
(There are no comments.)

RSC Subject Editor:
Comments to the Author:
(There are no comments.)

Reviewer(s)' Comments to Author:
Reviewer: 2

Comments to the Author(s)
I have carefully reviewed the responses, and my concerns have been adequately addressed.

Dra. Magdalena Malankowska
Chemical & Environmental Engineering
Department
University of Zaragoza
Escuela de Ingeniería y Arquitectura
C/ María de Luna, 3. 50018 Zaragoza. **SPAIN**
E-mail: magnal@unizar.es

Zaragoza, July 26th, 2019

RSC

Manuscript ID: RSOS-190866

Dear Editor,

Please find enclosed the revised **“Polyether-block-amide copolymer membrane for CO₂/N₂ separation: The influence of casting solution concentration on its morphology, thermal properties and gas separation performance”** by Lidia Martínez-Izquierdo, Magdalena Malankowska, Javier Sánchez-Laínez, Carlos Téllez and Joaquín Coronas for consideration for publication in Royal Society Open Science as full length article.

In the next pages of this letter you will find our answers to the point-by-point comments made by the reviewers. We highlighted in yellow the text changed or added to the manuscript.

Thanks in advance. I am looking forward to hearing from you soon.

Yours faithfully,

Magdalena Malankowska

RESPONSE TO REVIEWS:

-Reviewer 1:

This manuscript describes the fabrication of polyether-block-amide (PEBA) membranes for CO₂ separation. The authors prepared the PEBA membranes by varying the polymer concentration in the casting solution (1, 3, 5 wt.%) to investigate its influence on the morphology, thermal property, and gas separation performance. It was found that the amount of polymer in the casting solution affected the crystallinity and thermal stability of prepared membranes and the optimal content of PEBA was 3.0 wt.% because it showed the highest CO₂ permeability and CO₂/N₂ selectivity due to the lowest crystallinity without the PEBA chain entanglement. However, I cannot accept this manuscript because the effect of polymer concentration in the casting solution on the properties of the membrane has already been reported in many previous studies and any clear explanation on the separation behavior of the prepared membranes according to the mainly investigated research results (morphology, thermal property) is not possible. Therefore, this manuscript is not suited to be published in ‘Royal Society Open Science’.

Authors: We thank the reviewer for her/his effort revising our manuscript. As far as we know, there are not so many manuscripts in the open literature related to the effects that the polymer concentration in the casting solution has on the gas separation properties of membranes. The closest paper we have found in which different concentrations of PEBA have been tested was published by Ren et al.¹ In this article, they prepared solutions with different concentrations of PEBA (from 1 to 6 wt%) in ethanol/water, but the differences they found in the gas separation performance were mainly related to the thickness of the selective layer. In this case, the thickness changed depending on the PEBA concentration in the casting solution. However, although they have tested different concentrations, they have not reported the influences that this parameter has on the morphology and thermal properties of the membranes, as well as the relationship between these characteristics and the gas separation performance. We believe that the recent high

interest on Pebax 1657 membranes merits studies trying to gain insight into the physical properties of the polymer beyond its mere gas separation properties. This will make possible to better understand the separation ability of the polymer finding the proper correlation between polymer properties obtained at different casting conditions and the separation performance. To clarify this point, a new sentence has been added to the end of the Introduction as follows (Page 1):

“As far as we are concerned, while these aforementioned parameters have already been studied for PEBA,² there is no study about how the concentration in the casting solution affects the crystallinity and thermal properties of the polymer Pebax® MH 1657, although already studied by Ren et al.¹ for the gas separation performance. This is of great importance to find correlation between the casting conditions and the whole membrane performance, i.e. not only what limits the separation performance but what relates other issues such as morphology, preparation reproducibility, long period stability, etc. The changes in the concentration of polymer in the casting solution directly affects the viscosity, which will lead to differences in the behavior of membranes. As well as the solvent type, the viscosity will affect the evaporation time and this will result in differences in crystallinity and hence in permeability and selectivity, as occurred in the studies of Shao et al.³ and Karamouz et al.⁴ Therefore, with the aim of further optimization of the PEBA, in this work, we have carried out a study of the influence of the concentration of Pebax® MH 1657 in the casting solution on the corresponding gas separation membranes.”

-Reviewer 2

The present study reported the effect of polymer concentration on the morphology, thermal properties and CO₂/N₂ separation performance of PEBA membrane. The topic of CO₂ separation is of general interests to the membrane society, and PEBA membrane is a promising candidate for CO₂ separation. Overall the layout of the manuscript is all right, and some results might be useful even though the manuscript suffered from insufficient novelty.

A.: We thank the reviewer for dedicating time to evaluate our work. All the recommendations will be considered below.

1. Please specify which gas separation process was studied in the title.

A: We thank the reviewer for his/her statement. The gas separation mixture studied has been added to the title of the manuscript and it is marked in yellow.

2. Avoid using uncommon abbreviation of Pebax MH 1657 in the abstract. Full name is recommended.

A: We thank the reviewer for his/her suggestion. The full name is now provided in the abstract.

“The present work is focused on the study of the effect that the casting solution concentration has on the morphology and gas separation performance of poly(ether-block-amide) copolymer membranes (Pebax® MH 1657)”

3. A major weakness of the manuscript is that authors failed to justify the necessity to study the effect of polymer concentration, and missed other important preparation conditions, such as solvent type, solvent evaporation temperature.

A: We thank the reviewer for his/her statement. A paragraph justifying the necessity of this study has been added to the introduction. In addition, see also our answer to Referee #1.

Page 1 (Introduction section): “As far as we are concerned, while these aforementioned parameters have already been studied for PEBA,² there is no study about how the concentration in the casting solution affects the crystallinity and thermal properties of the polymer Pebax® MH 1657, although already studied by Ren et al.¹ for the gas separation performance. This is of great importance to find correlation between the casting conditions and the whole membrane performance, i.e. not only what limits the separation performance but what relates other issues such as morphology, preparation reproducibility, long period stability, etc. The changes in the concentration of polymer in the casting solution directly

affects the viscosity, which will lead to differences in the behavior of membranes. As well as the solvent type, the viscosity will affect the evaporation time and this will result in differences in crystallinity and hence in permeability and selectivity, as occurred in the studies of Shao et al.³ and Karamouz et al.⁴ Therefore, with the aim of further optimize the PEBA, in this work, we have carried out a study of the influence of the concentration of Pebax® MH 1657 in the casting solution on the corresponding gas separation membranes.”

4. Membrane preparation section: please indicate how much weight of solution was poured, and how to control the membrane thickness.

A: We thank the reviewer for his/her advice. A sentence in the membrane preparation section has been added in order to clarify how the control of the membrane thickness was achieved.

Page 2: “Note that the same amount of polymer (0.2 g) has been used to prepare the three casting solutions and that the total weight of each solution changes due to the different amount of solvent used to prepare them. The amount of polymer was fixed to obtain membranes of about the same thickness (40 μm) to be easily compared”

5. Some figure numbers in the manuscript were wrong.

A: We apologize to the reviewer for the mistake. Certainly, some figure numbers along the manuscript are wrong. We have correct these figures.

6. Authors are recommended to further explain why evaporation time will have an effect on the reorder of the polymer chain and why the chain order would affect the permeability.

A: We thank the reviewer for his/her suggestion. The influence of the evaporation time on the reorder of polymer chains and gas separation performance has been already explained more in detail in the reference provided in the text (ref.: 11): “Pebax membrane for CO₂/CH₄ separation: effects of various solvents on morphology and performance”. In this article, Isanejad et al., show how the use of different solvents affects the gas performance due to the changes in crystallinity.

Such differences of crystallinity are related to the evaporation time, which changes depending on the volatility of the solvent used.

The sentence has been changed in the article, making reference to the results of this study, in order to clarify and justify such statement.

Page 6: *“This effect of entanglement has been previously reported by Isanejad et al.² In their case, differences of crystallinity were related to the volatility of the solvent used to dissolve the PEBA. Such volatility implied that the time required to completely evaporate the solvent was distinct and so was the separation performance of membranes. In fact, they found that the most crystalline membranes were those with the highest selectivity and lowest permeability, in agreement with our findings.”*

- 7. Authors stated that “comparing the permeability and selectivity values presented in Table 2, ... through the membrane”. The comparison was unfair and the conclusion was not solid. Taking PEBA1 for example, the permeability increased 48%, whereas the selectivity decreased 52% when increasing temp. from 25 to 50C. The gas selectivity is more influenced than the permeability.**

A: We thank the reviewer for his/her suggestion. The text has been modified as follows:

Page 6: “Comparing the permeability and selectivity values presented in Table 2, it can be observed that both, gas selectivity and permeability, are similarly influenced by the changes in the PEBA concentration in the casting solution. On the other hand, these two parameters (selectivity and permeability) are also influenced by the operating temperature. Taking PEBA1 as an example, permeability increased up to 48 %, whereas the CO₂/N₂ selectivity decreased down to 52 %, when increasing the temperature from 25 to 50 °C.”

- 8. Error bar for the figure of CO₂/N₂ selectivity was missing.**

A.: We thank the reviewer for his/her advice. Error bars have been added to the CO₂/N₂ selectivity figure.

-Reviewer 3

The manuscript describes the effect of concentration of polymer solution on the morphology and gas permeation properties of PEBA membranes. Authors have studied three different concentrations and the membranes were characterized by SEM, XRD and thermal analysis. Authors use 1, 3, 5 wt% of polymer concentration. Membrane cast using 3wt% polymer shows optimum properties. The work is a systematic study and it will be a useful contribution to the literature in this area. The manuscript is recommended for publication in Royal Society Open Science after minor corrections are applied.

A: We thank the reviewer for dedicating time to evaluate our work. All the recommendations will be considered below.

Minor comments are following.

1. The figure numbers should be checked carefully.

A: We apologize to the reviewer for the mistake. Certainly, some figure numbers along the manuscript are wrong. We have correct these figures.

2. English proof reading is recommended.

A: Thanks for the comment. English has been revised by a highly proficient English speaker.

1. X. Ren, J. Ren, H. Li, S. Feng and M. Deng, *International Journal of Greenhouse Gas Control*, 2012, **8**, 111-120.
 2. M. Isanejad, N. Azizi and T. Mohammadi, *Journal of Applied Polymer Science*, 2017, **134**.
 3. L. Shao, T. Chung, G. Wensley, S. Goh and K. Pramoda, *Journal of Membrane Science*, 2004, **244**, 77-87.
 4. F. Karamouz, H. Maghsoudi and R. Yegani, *Journal of Natural Gas Science and Engineering*, 2016, **35**, 980-985.
-